# Neurological outcomes and predictive factors in traumatic spinal cord injury patients in the intensive care unit

Danqin Yuan[1☯], Yaling Jin[2,3☯], Lihui Chen[2,3☯], Min Tang[2,3], Jiuzhou Lin[2,3], Weiting Chen[2,3*]

1 Department of Emergency and Intensive Care Unit, Taizhou Integrated Chinese and Western Medicine Hospital, Taizhou, Zhejiang, China, 2 Department of Emergency and Intensive Care Unit, The First People's Hospital of Linhai, Taizhou, Zhejiang, China, 3 Department of Emergency and Intensive Care Unit, Linhai Branch of Second Affiliated Hospital, Zhejiang University School of Medicine, Taizhou, Zhejiang, China

☯ These authors contributed to the work equally.
* chenweitingwl@sina.com

## Abstract

### Background

This study aims to analyze the demographic characteristics, clinical features, and neurological outcomes of patients with traumatic spinal cord injury (TSCI) admitted to the Intensive Care Unit (ICU), in order to provide scientific evidence for the prevention and management of TSCI.

### Methods

A retrospective analysis was conducted on data from TSCI patients admitted to the ICU between January 2018 and December 2022. Demographic information, neurological injury characteristics, complications during hospitalization, treatment interventions, and prognosis were comprehensively collected. Based on changes in neurological function before and after treatment, patients undergoing surgery were classified into improvement and non-improvement groups. Neurological recovery was assessed using the American Spinal Injury Association (ASIA) impairment scale. Univariate and multivariate logistic regression analyses were performed to identify key factors influencing neurological recovery.

### Results

A total of 341 TSCI patients were included, with a mean age of 55.2±13.4 years and a male-to-female ratio of 6.3:1. The leading cause of TSCI were high falls (47.5%), traffic accidents (35.8%) and low falls (9.1%). Cervical spinal cord was most common, followed by thoracic and lumbar cord. Among surgical patients, the neurological improvement rate was 14.8%, compared to 12.5% in non-surgical

**Data availability statement:** All relevant data are within the manuscript.

**Funding:** The author(s) received no specific funding for this work.

**Competing interests:** The authors have declared that no competing interests exist.

patients, highlighting the potential benefits of surgical intervention. Multivariate analysis revealed that early targeted blood pressure management (MAP ≥ 85 mmHg) (OR=2.296, 95% CI: 1.036–5.086, P = 0.040) and early surgery (≤ 24h) (OR=2.841, 95% CI: 1.088–7.419, P = 0.033) were significant protective factors for neurological improvement.

## Conclusions

Patients with TSCI admitted to the ICU are predominantly middle-aged men, with high falls and traffic accidents being the primary causes. Early blood pressure optimization and timely surgical intervention are significantly associated with improved neurological outcomes.

---

## Introduction

Traumatic spinal cord injury (TSCI) is a severe and potentially life-threatening that often results in significant disability and long-term complications [1–3]. The global annual incidence of TSCI is estimated to range between 10.4 to 83.0 cases per million, with reported rates of 27–83 in the United States and 21 to 32.3 in Australia [4–6]. In China, the incidence of TSCI increased markedly from 45.1 to 66.5 cases per million between 2009 and 2018 [7], underscoring its growing burden. As a highly prevalent traumatic condition, TSCI severely compromises quality of life [8], often requiring lifelong medical care and rehabilitation. This imposes substantial emotional and financial burdens on affected individuals, their families, and healthcare systems [9–11]. Annual medical expenditures for TSCI are estimated at approximately USD 30,770–62,653 per patient [10].

Despite recent advances in clinical care, effective treatment options for TSCI remain limited, making prevention a key strategy [12]. A comprehensive understanding of the incidence and current treatment landscape of TSCI is essential for developing targeted prevention and therapeutic interventions. Epidemiological studies provide valuable insights into patient profiles, injury mechanisms, and treatment trends, which can inform healthcare resource allocation, policy development, and clinical guidelines. Given the wide regional variation in TSCI incidence and risk factors, region-specific, in-depth research is essential. Such data can enhance risk stratification and prevention efforts, ultimately improving patient outcomes and public health responses.

Although some progress has been made in TSCI research, significant knowledge gaps persist, particularly regarding intensive care unit (ICU)-admitted TSCI. These patients are typically in critical condition, with a higher incidence of comorbid injuries, complications, and mortality. However, it remains unclear whether non-specialized ICU care adversely affects neurological recovery [13–15]. Studies specifically examining ICU-admitted TSCI patients remain scarce, often featuring small sample sizes (48–108 cases) and focusing mainly on cervical injuries, without comprehensive analysis of treatment strategies and neurological outcomes [14,15]. Furthermore, the impact of the coronavirus disease 2019 (COVID-19) pandemic on the epidemiological trends of TSCI

has not been fully elucidated. This study retrospectively analyzed hospital data to investigate the demographics and clinical characteristics, as well as neurological outcomes of ICU-admitted TSCI patients. The goal is to inform prevention and treatment strategies, provide a theoretical foundation for resource allocation, and ultimately enhance the effectiveness of clinical and public health responses to TSCI.

## Materials and methods

### Studying setting

This retrospective study reviewed medical records from the ICU of The Second Affiliated Hospital Zhejiang University School of Medicine. Patients admitted between January 2018 and December 2022 were identified using the International Classification of Disease, 10th Revision (ICD-10), and the diagnosis code for TSCI. Data were accessed and analyzed between December 4, 2023, and December 4, 2024. The final diagnosis was determined based on the patient's diagnosis at discharge or death.

### Participants

Three investigators independently reviewed the medical records of patients with TSCI admitted to the ICU between January 1, 2018, and December 7, 2022. Inclusion criteria were: (1) adult patients (≥18 years), (2) trauma-induced spinal cord injury, and (3) ICU admission at the time of injury. Exclusion criteria included: non-TSCI; age < 18 years; intervertebral disc disease, spinal fractures without spinal cord involvement, preexisting neuromuscular or psychiatric disorders; craniocerebral injuries precluding cooperation; incomplete or unclear medical records; Grade E of the American Spinal Injury Association (ASIA) Impairment Scale; and patients with fatal injuries who were not hospitalized. Patients with ASIA Grade E were excluded due to the absence of neurological deficits and the minimal risk of functional deterioration.

### Data collection

The following variables were collected: age, gender, body mass index (BMI), occupation, marital status, comorbidities, education level, time of injury, cause of injury, level of injury, ASIA Impairment Scale grade, severity of injury, Injury Severity Score (ISS), surgical intervention and approach, injury segments, ICU and in-hospital length of stay (LOS), associated injuries, and complications. Individual participants were identifiable during or after data collection.

Based on the timeline of COVID-19 public health measures, patients were categorized into pre-pandemic control (before January 21, 2020) and post- pandemic control (January 21, 2020, to December 7, 2022) groups. patients were stratified into six age groups: 18–30, 31–45, 46–60, 61–75, 76 and above [16]. Marital status was categorized as married, unmarried, divorced, and widowed. The causes of injuries were classified as traffic accidents, low falls (<1 m), high falls (≥ 1 m), injuries from falling objects, assaults, and sports-related injuries [17]. ASIA grades (A to E) were used to assess sensory and motor function below the level of injury [18]. Associated injuries included spinal fracture and/or dislocation, craniocerebral trauma, maxillofacial trauma, thoracic and abdominal injuries, pelvic fracture, and limb fracture and/or dislocation. Any of such injury was considered an associated injury. Complications include acute respiratory failure (ARF), pneumonia, deep vein thrombosis (DVT), pulmonary embolism (PE), and cardiac arrest. Any of these was considered TSCI-related. Early surgery was defined as surgery intervention within 24 hours injury [19]. Early target blood pressure management was defined as maintaining mean arterial pressure (MAP) ≥ 85 mmHg within the first 3 days after injury [12]. According to pharmaceutical guidelines, patients receiving methylprednisolone sodium succinate or methylprednisolone are divided into "high-dose" (≥ 500 mg) and "regular-dose" (< 500 mg) groups.

### Ethical considerations

This study was approved by the Ethics Committee of the Second Affiliated Hospital Zhejiang University School of Medicine (Approval No. 2023-0287).

## Statistical analysis

Descriptive statistics were used to summarize the demographic and clinical characteristics of patients. Data with normal distribution were presented as mean ± standard deviation (SD), while non-normal distribution data were expressed as median with interquartile range (IQR). Categorical variables were reported as frequencies and percentages. A P-value < 0.05 was considered statistically significant. Univariate and multivariate logistic regression analyses were conducted to identify significant predictors of neurological improvement. Multivariate models adjusted for age, gender, ISS, the level and severity of the injury, steroid treatment, the use of mannitol, associated injuries and surgical approach. All statistical analyses were conducted using SPSS for Mac package version 21.0 (SPSS Inc., Chicago) and Microsoft Excel (Microsoft Corporation, Redmond, WA, USA). Graphs were generated using GraphPad Prism7 (GraphPad Software, CA, USA).

## Results

### Patient selection

Between January 1, 2018, and December 7, 2022, a total of 491 TSCI cases were identified. Of these, 150 cases were excluded for the following reasons: readmission (n = 34), missing data (n = 49), non-traumatic or unclear diagnosis (n = 22), severe craniocerebral injury leading to no-cooperation (n = 33), and psychiatric disorders, ASIA Grade E, or age < 18 years (n = 12). Ultimately, 341 patients confirmed TSCI admitted to the ICU were included in the analysis (Fig 1).

### General characteristics of TSCI patients

Table 1 summarizes the demographic characteristics of the TSCI patients in the pre-pandemic control group (30.8%, 105/341) and the post-pandemic control group (69.2%, 236/341). The age of the patients ranged from 16 to 84 years, with a mean age of 55.2 ± 13.4 years. The average age was higher in the post-pandemic control group (56.0 years) compared to the pre-pandemic control group (53.3 years), although the difference was not statistically significant (P > 0.05). The age group 46–60 years represented the highest proportion in both groups. Among the 341 patients, 294 (86.2%) were male and 47 (13.8%) females, resulting in a male-to-female ratio of 6.3:1. The male predominance remained consistent across both groups. No statistically significant difference in gender distribution was observed (P > 0.05).

Regarding educational level, most patients had completed elementary or middle school (76.2% in the pre-pandemic group and 74.2% in the post-pandemic groups). There were no significant differences between the groups (P > 0.05). In terms of occupation, peasants and laborers were the most represented groups. In the pre-pandemic group, peasants accounted for 37.1% and laborers 34.3%. Similarly, in the post-pandemic group, peasants made up 38.6% and laborers 36.9%. These findings suggest that individuals in physically demanding occupations are at higher risk for TSCI.

### Etiology of injury

The etiologies of TSCI are presented in Fig 2. The leading cause was high falls (47.5%), followed by traffic accidents (35.8%) and low falls (9.1%). In the post-pandemic control group, high falls remained the most common cause (47.6%), followed by traffic accidents (41.9%), and injuries from falling objects (5.7%) (Fig 2A). In the pre-pandemic control group, high falls (47.5%) were also predominant, followed by traffic accidents (33.1%) and low falls (11.9%) (Fig 2C). Compared to the pre-control group, the incidence of traffic accidents decreased in the post-control group, whereas low falls increased significantly (P < 0.05).

Further analysis of traffic accident-related injuries revealed that in the post-control group, two-wheeled vehicles were responsible for 50.0% of cases, followed by four-wheeled vehicles (15.9%) and three-wheeled vehicles (13.6%) (Fig 2B). In the pre-pandemic group, two-wheeled vehicles were also the leading cause (59.0%), followed by four-wheeled vehicles

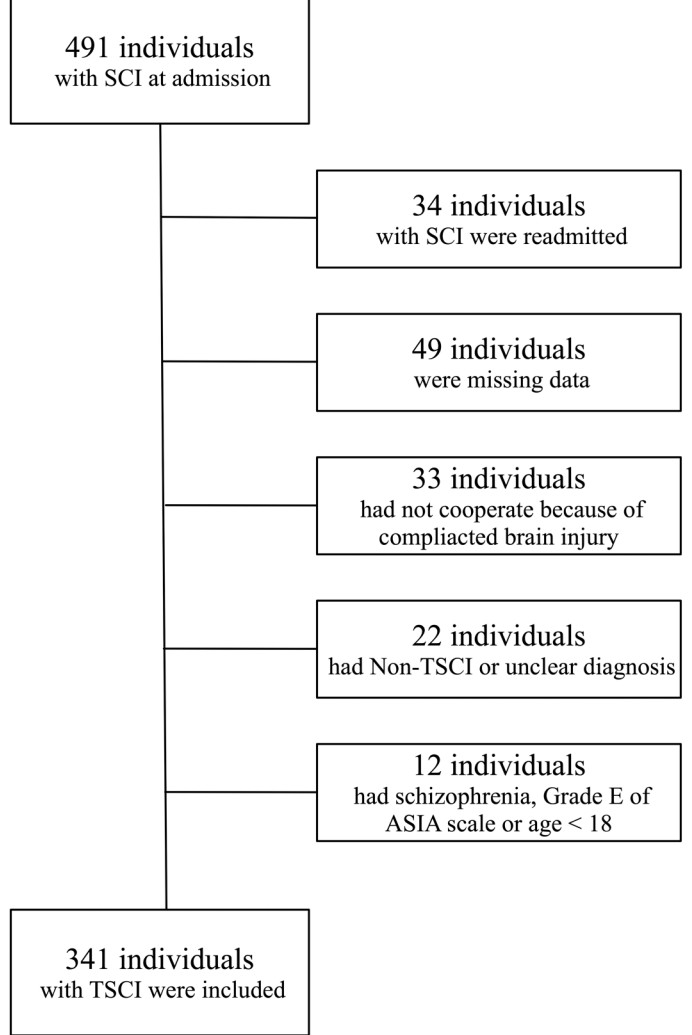

**Fig 1. Flow diagram of individuals included with TSCI.**

(10.3%) and bicycles (10.3%) (Fig 2D). Although the proportion of injuries due to four-wheeled vehicles decreased while two-wheeled vehicles and bicycles increased, these differences were not statistically significant (P > 0.05).

Table 2 presents injury etiology by gender. Among male patients, high falls (49.7%) were the most frequent cause, while traffic accidents were the most common among female patients (44.7%). Table 3 shows age-stratified etiologies. Traffic accidents and high falls were the leading causes in the 46–60 age group, with statistically significant differences observed across age groups (P < 0.05).

## Level of injury

Fig 3 illustrates a bimodal distribution of injury levels. The cervical spine cord was the frequently affected region (83.5% of all cases), particularly segments C3–C6, which accounted for 73.3% of cervical injuries. The second most affected region was the thoracolumbar junction, especially T11–L1 (7.3%). The most frequently injured vertebrae were C4 (28.6%), T11 (23.7%), and L1 (45.0%) within the cervical, thoracic, and lumbar regions, respectively.

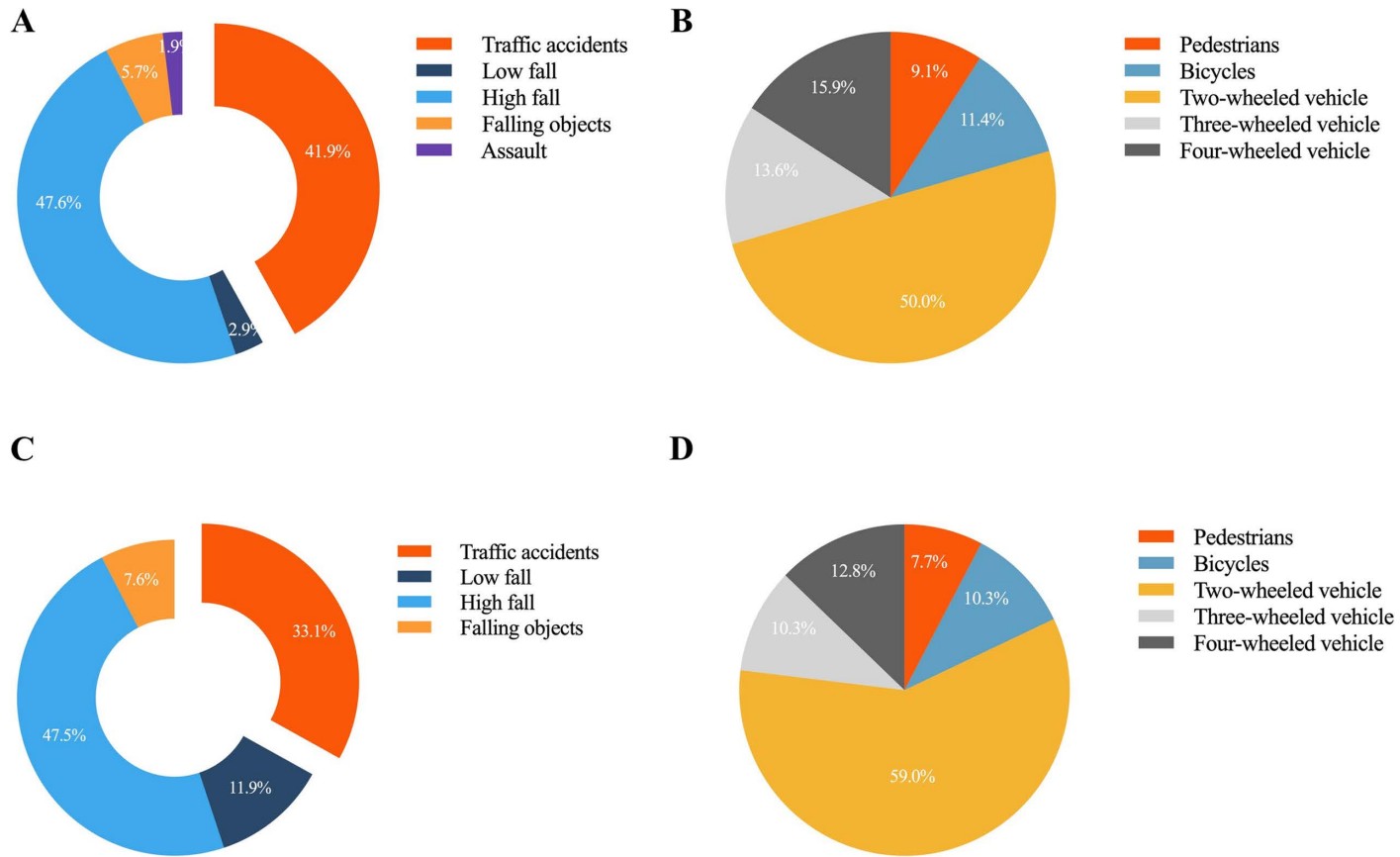

**Fig 2. The etiology of TSCI (A/C) and details of traffic accidents (B/D) in pre- and post-pandemic control.**

In the pre-pandemic group, cervical, thoracic, and lumbar injuries accounted for 88.6%, 8.6% and 2.9% of cases, respectively. In the post-pandemic group, these figures were 80.1%, 12.7%, and 7.2%, respectively. Although the proportion of thoracic and lumbar injuries increased slightly in the post-pandemic group, the differences were not statistically significant (P > 0.05).

### ASIA grade and injury severity

According to the ASIA Impairment Scale, the distribution of neurological impairments was as follows: Grade A (30.8%, 105/341), Grade B (2.6%, 9/341), Grade C (24.6%, 84/341), and Grade D (41.9%, 143/341) (Fig 4A). Although the proportion of Grade A injuries appeared to increase in the post-pandemic control group, the difference was not statistically significant (P > 0.05).

In terms of injury severity, patients were categorized as having complete quadriplegia, incomplete quadriplegia, complete paraplegia, and incomplete paraplegia. Most cases were incomplete quadriplegia (60.7%), followed by complete quadriplegia (22.0%), complete paraplegia (10.9%), and incomplete paraplegia (6.5%) (Fig 4B). While the proportion of complete paraplegia increased slightly in the post-pandemic control group, no statistically significant difference was observed (P > 0.05).

Table 4 presents AISA grades by injury level. Most cervical cord injuries were classified as Grade D (38.1%) or Grade C (22.3%). In contrast, thoracic injuries were most frequently classified as Grade A (9.1%).

**Table 1. Demographic and clinical characteristics of 341 TSCI patients in pre- and post-pandemic control.**

| Characters | Total N = 341 (%) | Pre-pandemic control N = 105 (%) | Post-pandemic control N = 236 (%) | P value |
|---|---|---|---|---|
| Age (year) | | | | 0.318 |
| 16-30 | 19 (5.6) | 7 (6.7) | 12 (5.1) | |
| 31-45 | 49 (14.4) | 14 (13.3) | 35 (14.9) | |
| 46-60 | 143 (42.1) | 52 (49.5) | 91 (38.7) | |
| 61-75 | 114 (33.5) | 28 (26.7) | 86 (36.6) | |
| >75 | 15 (4.4) | 4 (3.8) | 11 (4.7) | |
| Age (year) | 55.2 ± 13.4 | 53.3 ± 12.6 | 56.0 ± 13.7 | 0.083 |
| BMI (kg/m$^2$) | 22.9 (20.7, 25.0) | 22.2 (20.3, 24.5) | 23.1 (21.0, 25.1) | 0.006 |
| APACHE II | 8.0 (6.0, 11.0) | 8.0 (5.0, 11.0) | 8.0 (6.0, 11.0) | 0.792 |
| Smoking history | | | | 0.752 |
| Yes | 184 (54) | 58 (55.2) | 126 (53.4) | |
| No | 157 (46) | 47 (44.8) | 110 (46.6) | |
| Gender | | | | 0.390 |
| Male | 294 (86.2) | 88 (83.8) | 206 (87.3) | |
| Female | 47 (13.8) | 17 (16.2) | 30 (12.7) | |
| Underlying diseases | | | | 0.264 |
| Yes | 122 (35.8) | 33 (31.4) | 89 (37.7) | |
| No | 219 (64.2) | 72 (68.6) | 147 (62.3) | |
| *Marital status* | | | | 0.964 |
| Unmarried | 18 (5.3) | 5 (4.8) | 13 (5.5) | |
| Married | 308 (90.3) | 95 (90.5) | 213 (90.2) | |
| Divorced | 10 (2.9) | 3 (2.8) | 7 (3.0) | |
| Widowed | 5 (1.5) | 2 (1.9) | 3 (1.3) | |
| *Education level* | | | | 0.801 |
| Illiterate | 23 (6.7) | 6 (5.7) | 17 (7.2) | |
| Elementary school | 124 (36.4) | 36 (34.3) | 88 (37.3) | |
| Middle school | 131 (38.4) | 44 (41.9) | 87 (36.9) | |
| High school | 35 (10.3) | 9 (8.6) | 26 (11.0) | |
| College or more | 28 (8.2) | 10 (9.5) | 18 (7.6) | |
| *Occupation* | | | | 0.557 |
| Peasant | 130 (38.1) | 39 (37.1) | 91 (38.6) | |
| Laborer | 123 (36.1) | 36 (34.3) | 87 (36.9) | |
| Drivers | 9 (2.6) | 5 (4.8) | 4 (1.7) | |
| Worker | 16 (4.7) | 3 (2.9) | 13 (5.5) | |
| Students | 4 (1.2) | 1 (1.0) | 3 (1.3) | |
| Retired | 28 (8.2) | 8 (7.6) | 20 (8.5) | |
| Teachers | 2 (0.6) | 1 (1.0) | 1 (0.4) | |
| Others* | 29 (8.5) | 12 (11.4) | 17 (7.2) | |
| *Etiology* | | | | 0.012 |
| Traffic accidents | 122 (35.8) | 44 (41.9) | 78 (33.1) | |
| Low fall (<1m) | 31 (9.1) | 3 (2.9) | 28 (11.9) | |
| High fall (≥ 1m) | 162 (47.5) | 50 (47.6) | 112 (47.5) | |
| Falling objects | 24 (7.0) | 6 (5.7) | 18 (7.6) | |

*(Continued)*

**Table 1.** (Continued)

| Characters | Total<br>N = 341 (%) | Pre-pandemic control<br>N = 105 (%) | Post-pandemic control<br>N = 236 (%) | P value |
|---|---|---|---|---|
| Assault | 2 (0.6) | 2 (1.9) | 0 (0) | |
| Level of the injury | | | | 0.131 |
| Cervical | 282 (82.7) | 93 (88.6) | 189 (80.1) | |
| Thoracic | 39 (11.4) | 9 (8.6) | 30 (12.7) | |
| Lumbar | 20 (5.9) | 3 (2.9) | 17 (7.2) | |
| Sacral | 0 (0) | 0 (0) | 0 (0) | |
| ASIA grade | | | | 0.646 |
| A | 105 (30.8) | 29 (27.6) | 76 (32.2) | |
| B | 9 (2.6) | 2 (1.9) | 7 (3.0) | |
| C | 84 (24.6) | 25 (23.8) | 59 (25.0) | |
| D | 143 (41.9) | 49 (46.7) | 94 (39.8) | |
| The level and severity of the injury | | | | 0.262 |
| Complete tetraplegia | 75 (22.0) | 23 (21.9) | 52 (22.0) | |
| Incomplete tetraplegia | 207 (60.7) | 70 (66.7) | 137 (58.1) | |
| Complete paraplegia | 37 (10.9) | 8 (7.6) | 29 (12.3) | |
| Incomplete paraplegia | 22 (6.5) | 4 (3.8) | 18 (7.6) | |
| ISS | 25.0 (18.0, 33.0) | 25.0 (20.0, 30.0) | 26.0 (18.0, 33.8) | 0.215 |
| Treatment strategy | | | | 0.698 |
| Surgical treatment | 277 (81.2) | 84 (80.0) | 193 (81.8) | |
| Conservative treatment | 64 (18.8) | 21 (20.0) | 43 (18.2) | |
| Mechanical ventilation | | | | 0.251 |
| Yes | 179 (52.5) | 60 (57.1) | 119 (50.4) | |
| No | 162 (47.5) | 45 (42.9) | 117 (49.6) | |
| Tracheotomy | | | | <0.001 |
| Yes | 107 (31.4) | 48 (45.7) | 59 (25.0) | |
| No | 234 (68.6) | 57 (54.3) | 177 (75.0) | |
| Combined injury | | | | |
| Fracture dislocation of spine | 324 (95.0) | 99 (94.3) | 225 (95.3) | 0.680 |
| Craniocerebral injury | 105 (30.8) | 34 (32.4) | 71 (30.1) | 0.672 |
| Maxillofacial injuries | 70 (20.5) | 14 (13.3) | 56 (23.7) | 0.028 |
| Chest trauma | 187 (54.8) | 49 (46.7) | 138 (58.5) | 0.043 |
| Abdominal | 19 (5.6) | 4 (3.8) | 15 (6.4) | 0.344 |
| Pelvic fracture | 8 (2.4) | 4 (3.8) | 4 (1.7) | 0.234 |
| Limb fracture | 60 (17.6) | 19 (18.1) | 41 (17.4) | 0.872 |
| Complication | | | | |
| ARF | 183 (53.7) | 62 (59.0) | 121 (51.3) | 0.184 |
| Pneumonia | 169 (49.6) | 60 (57.1) | 109 (46.2) | 0.062 |
| DVT | 82 (24.1) | 12 (11.4) | 70 (29.7) | 0.000 |
| PE | 15 (4.4) | 3 (2.9) | 12 (5.1) | 0.354 |
| Cardia arrest | 3 (0.9) | 2 (1.9) | 1 (0.4) | 0.176 |
| Physical VTE prophylaxis n (%) | | | | 0.941 |
| Yes | 293 (85.9) | 90 (85.7) | 203 (86.0) | |
| No | 48 (14.1) | 15 (14.3) | 33 (14.0) | |

*(Continued)*

 

**Table 1.** (Continued)

| Characters | Total<br>N = 341 (%) | Pre-pandemic<br>control<br>N = 105 (%) | Post-pandemic<br>control<br>N = 236 (%) | P value |
|---|---|---|---|---|
| Chemical VTE prophylaxis n (%) | | | | 0.005 |
| Yes | 159 (46.6) | 61 (58.1) | 98 (41.6) | |
| No | 182 (53.4) | 44 (41.9) | 138 (58.4) | |
| ICU LOS in days, median (IQR) | 6.0 (3.0, 12.0) | 11.0 (3.0, 17.5) | 5.0 (3.0, 9.0) | <0.001 |
| In-hospital LOS in days, median (IQR) | 12.0 (8.0, 17.0) | 16.0 (11.5, 21.0) | 10.0 (7.0, 14.0) | <0.001 |

**Table 2. Distribution of etiology of injury by gender.**

| Gender | Etiology of the injury | | | | | |
|---|---|---|---|---|---|---|
| | Traffic accidents n (%) | Low fall n (%) | High fall n (%) | Falling objects n (%) | Assault n (%) | Total |
| Male | 101 (34.4) | 23 (7.8) | 146 (49.7) | 22 (7.5) | 2 (0.7) | 294 (100) |
| Female | 21 (44.7) | 8 (17.0) | 16 (34.0) | 2 (4.3) | 0 (0) | 47 (100) |
| Total | 122 (35.8) | 31 (9.1) | 162 (47.5) | 24 (7.0) | 2 (0.6) | 341 (100) |

**Table 3. Analysis of the etiologies and age distribution among patients with TSCI.**

| Etiologies | Age | | | | |
|---|---|---|---|---|---|
| | 16-30 n (%) | 31-45 n (%) | 46-60 n (%) | 61-75 n (%) | >75 n (%) |
| Traffic accident | 5 (1.5) | 19 (5.6) | 48 (14.1) | 47 (13.8) | 3 (0.9) |
| Low fall | 0 (0) | 3 (0.9) | 6 (1.8) | 14 (4.1) | 8 (2.3) |
| High fall | 11 (3.2) | 23 (6.7) | 76 (22.3) | 48 (14.1) | 4 (1.2) |
| Falling objects | 4 (1.2) | 4 (1.2) | 11 (3.2) | 5 (1.5) | 0 (0) |
| Assault | 0 (0) | 0 (0) | 2 (0.6) | 0 (0) | 0 (0) |

## Combined injury

Among the 341 patients, 335 (98.2%) sustained one or more associated injuries. The most common was spinal fracture and/or dislocation (94.9%), followed by chest trauma (53.8%), craniocerebral injury (30.8%), maxillofacial injuries (19.5%), limb fracture (16.5%), abdominal injury (6.2%), pelvic fractures (2.4%). Compared to the pre-pandemic control group, the incidence of maxillofacial injuries and chest trauma increased significantly in the post-pandemic group (13.3% vs. 23.7% and 46.7% vs 58.5%, respectively; both P < 0.05) (Fig 5).

## Complications

During hospitalization, 250 patients (73.3%) developed at least one complication. The most frequent was ARF (53.7%), followed by pneumonia (49.6%), DVT (24.1%), PE (4.4%) and cardiac arrest (0.9%). The incidence of DVT increased significantly in the post-pandemic group (29.7%) compared to the pre-pandemic group (11.4%) (P < 0.05) (Fig 6).

Mechanical ventilation was required in 52.5% of patients due to ARF, and 31.4% underwent tracheotomy. There was no significant difference in the use of mechanical ventilation between groups (P > 0.05). However, tracheotomy rates were significantly lower in the post-pandemic group than in the pre-pandemic group (45.7% vs. 25.0%, P < 0.05).

## Treatment of TSCI and status at discharge

During hospitalization, treatment strategies were classified as either conservative or surgical. Conservative treatment consisted of dehydration therapy with mannitol, corticosteroids to reduce edema, and neuroprotective agents. Surgical interventions included spinal canal decompression, discectomy, bone graft fusion, and internal fixation.

 

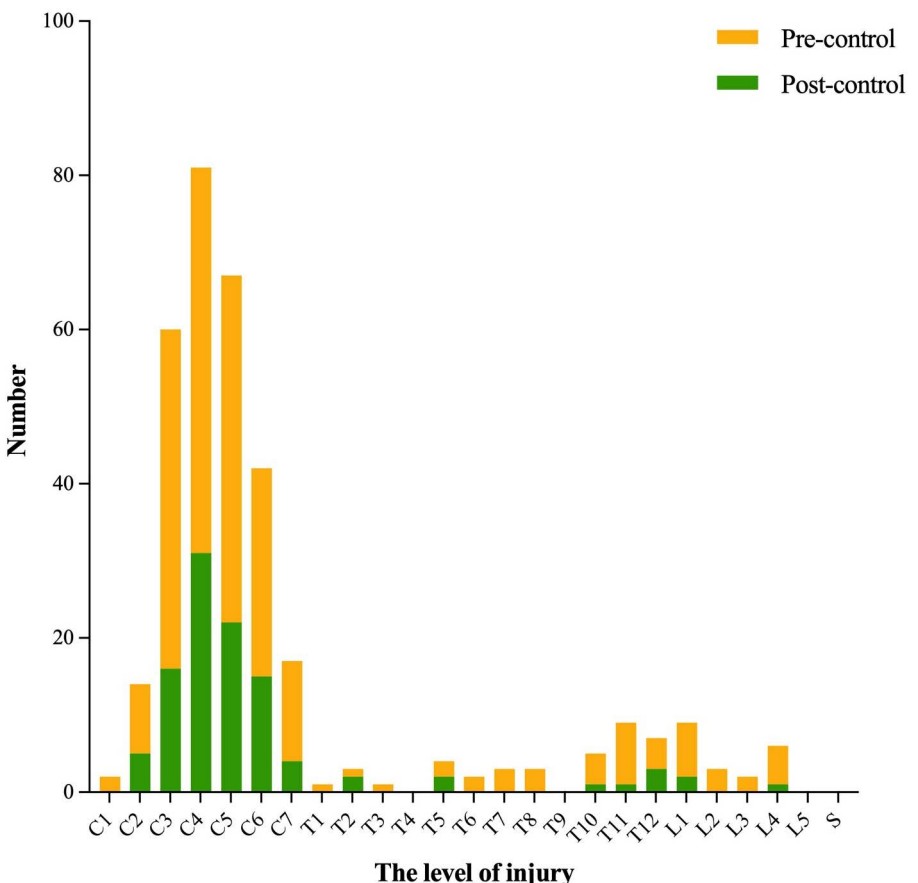

**Fig 3. Distribution histogram of the injury level of the patients.**

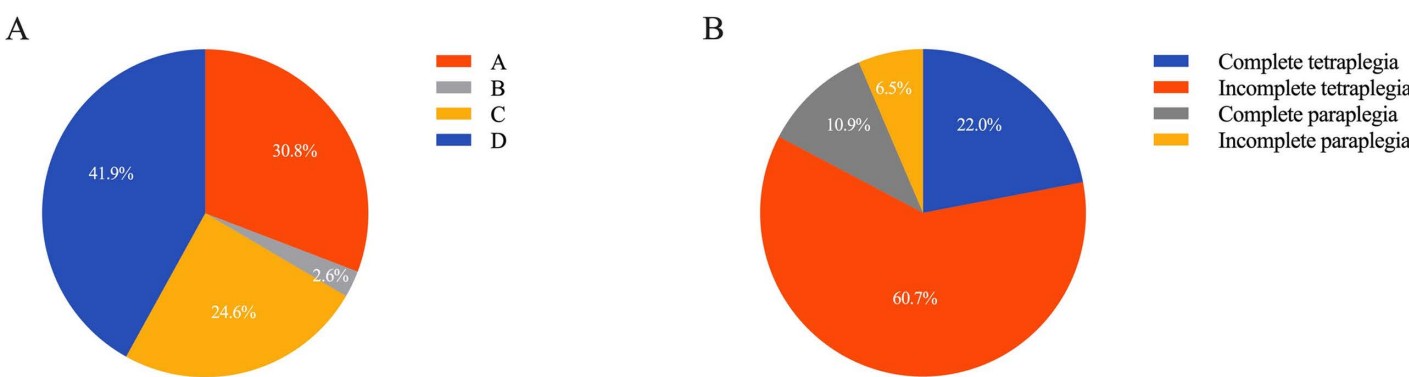

**Fig 4. The severity of patients with TSCI: (A) The degree of injury; (B) Their spinal function after injury evaluated by ASIA impairment scale.**

**Table 4. Comparison of the grade and segments of the injury among patients with TSCI.**

| ASIA scale | The level of injury | | | |
|---|---|---|---|---|
| | Cervical cord n (%) | Thoracic cord n (%) | Lumbar cord n (%) | Sacral cord n (%) |
| A | 69 (20.2) | 31 (9.1) | 5 (1.5) | 0 (0) |
| B | 7 (2.1) | 1 (0.3) | 1 (0.3) | 0 (0) |
| C | 76 (22.3) | 1 (0.3) | 7 (2.1) | 0 (0) |
| D | 130 (38.1) | 6 (1.8) | 7 (2.1) | 0 (0) |

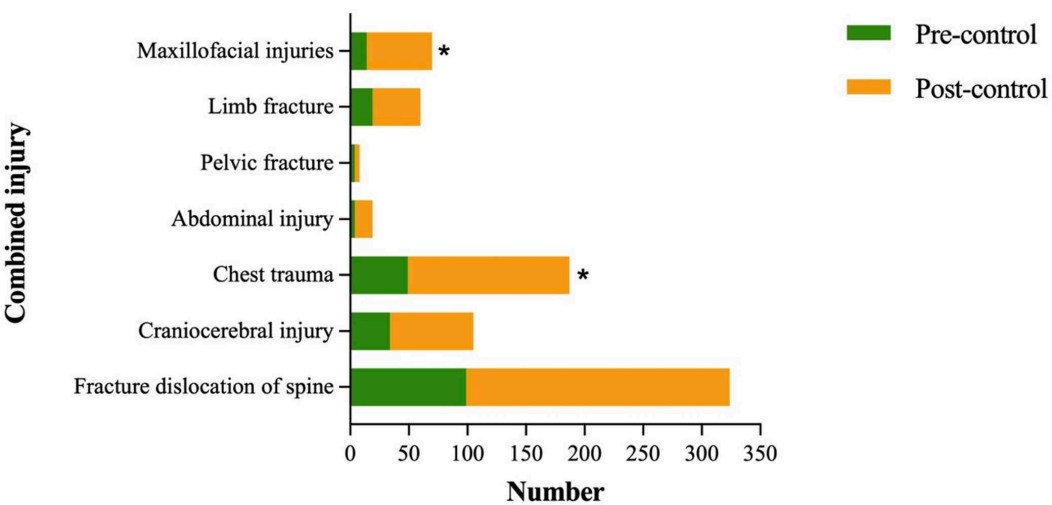

**Fig 5. Details of TSCI patients with combined injuries.** *Compared with pre-pandemic control, P<0.05.

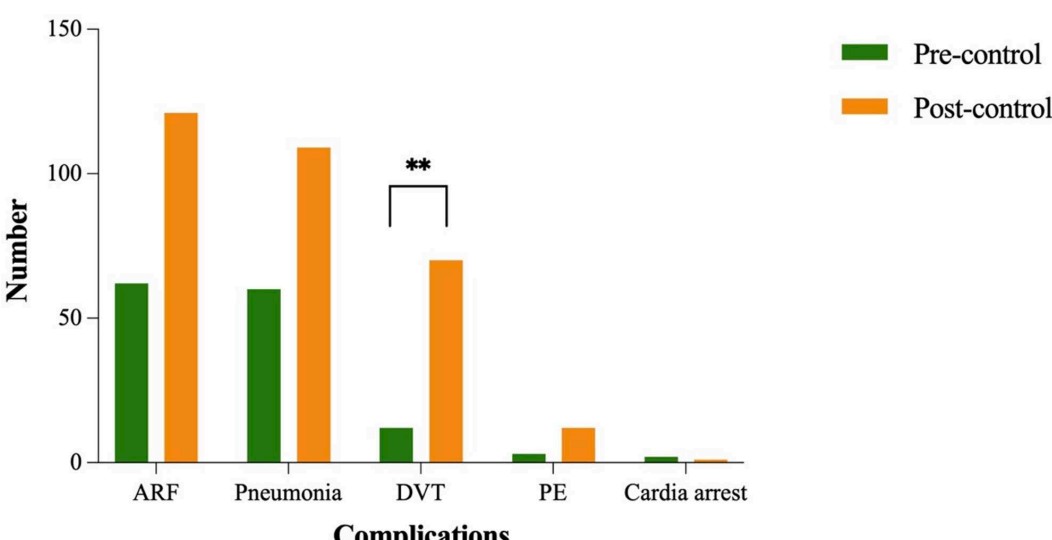

**Fig 6. The complications of TSCI before and after the control.** **compared with pre-pandemic control, P<0.001.

In total, 277 patients (81.2%) received surgical treatment, while 64 patients (18.8%) received conservative treatment, resulting in a surgical-to-conservative treatment ratio of 4.3:1. The mean preoperative interval was 6.6±4.7 days. Notably, the time to surgery was significantly shorter in the post-pandemic control group compared to the pre-pandemic control group (6.2±4.8 days vs 7.6±4.5 days, P<0.05). Patient outcomes after surgery or conservative treatment were analyzed as a whole [20]. As summarized in Table 5, ASIA grade changes from admission to discharge showed that 71.3% of patients experienced no neurological change, with similar proportions in both the surgical (71.1%) and conservative (71.9%) groups. Among patients who showed neurological improvement but remained at an incomplete injury level (ASIA Grade B/C/D), the improvement rate was 13.0% for surgical and 9.4% for conservative treatment. The cure rate, defined as discharge with ASIA Grade E, was low in both groups (0.9%). A proportion of patients experienced neurological deterioration, with deterioration rates of 15.2% and 17.2% in the surgical and conservative groups, respectively. No statistically significant difference in outcomes was observed between the two treatment modalities (P>0.05). Importantly, no in-hospital deaths occurred.

ICU and in-hospital length of stay (LOS) were significantly shorter in the post-pandemic group compared to the pre-pandemic control group (ICU LOS: 5.0 vs. 11.0 days, hospital LOS: 10.0 vs. 14.0 days; both P<0.001).

### Predictors of neurological improvement

To identify key predictors of postoperative neurological recovery, surgical patients were divided into improvement and non-improvement groups based on changes in ASIA grades between admission and discharge. Univariate analysis revealed that early surgery (≤ 24h) and early targeted blood pressure management (MAP ≥ 85 mmHg within 3 days) were significantly associated with neurological improvement (P<0.05) (Table 6).

Multivariate logistic regression analysis adjusting for age, gender, ISS, the level and severity of the injury, steroid use, mannitol administration, combined injuries and surgical approach confirmed that early targeted blood pressure management (OR=2.296, 95% CI: 1.036–5.086, P=0.040) and early surgery (OR=2.841, 95% CI: 1.088–7.419, P=0.033) were independent predictors for neurological improvement (Table 7). These findings underscore the critical role of timely hemodynamic stabilization and surgical intervention in optimizing neurological outcomes for TSCI patients.

## Discussion

This retrospective study analyzed the characteristics of patients with TSCI admitted to the ICU in Hangzhou, China, from 2018 to 2022. To the best of our knowledge, this is the first large-scale epidemiological investigation of TSCI in this region. Despite the implementation of COVID-19-related public health measures during the study period, our findings suggest a continued upward trend in TSCI incidence.

Our findings revealed a striking male predominance, with a male-to-female ratio of approximately 6.3:1, substantially higher than figures reported in prior studies [17,20,21]. This likely reflects the overrepresentation of males in high-risk occupations such as construction and manual labor, which are common sources of TSCI in China. Men are also more likely to engage in physically demanding and hazardous activities, increasing their vulnerability to severe trauma.

Table 5. Treatment of TSCI and functional changes in admission and discharge.

| Treatment of TSCI | Status on discharge | | | | |
|---|---|---|---|---|---|
| | Cure[a] n (%) | Improvement[b] n (%) | Unchanged[c] n (%) | Deterioration[d] n (%) | Total |
| Operative treatment | 2 (0.7) | 36 (13.0) | 197 (71.1) | 42 (15.2) | 277 (100) |
| Conservative treatment | 1 (1.6) | 6 (9.4) | 46 (71.9) | 11 (17.2) | 64 (100) |
| Total | 3 (0.9) | 42 (12.3) | 243 (71.3) | 53 (15.5) | 341 (100) |

**Table 6. Univariate analysis of postoperative neurological improvement in TSCI patients.**

| Characters | Improvement Group N = 39 (14.1%) | Non-Improvement Group N = 238 (85.9%) | P Value |
|---|---|---|---|
| Age (year) | | | 0.865 |
| 18-30 | 2 (5.1) | 16 (6.7) | |
| 31-45 | 7 (17.9) | 39 (16.4) | |
| 46-60 | 14 (35.9) | 104 (43.7) | |
| 61-75 | 14 (35.9) | 70 (29.4) | |
| >75 | 2 (5.1) | 9 (3.8) | |
| Gender | | | 0.067 |
| Male | 30 (76.9) | 209 (87.8) | |
| Female | 9 (23.1) | 29 (12.2) | |
| Smoking history | | | 0.835 |
| Yes | 22 (56.4) | 130 (54.6) | |
| No | 17 (43.6) | 108 (45.4) | |
| Underlying diseases | | | |
| Hypertension | 11 (17.7) | 51 (82.3) | 0.347 |
| Diabetes | 4 (21.1) | 15 (78.9) | 0.365 |
| Cardiovascular disease | 2 (33.3) | 4 (66.7) | 0.170 |
| Respiratory diseases | 1 (20.0) | 4 (80.0) | 0.701 |
| Etiology | | | 0.746 |
| Traffic accidents | 18 (46.2) | 85 (35.7) | |
| Low fall (<1m) | 3 (7.7) | 21 (8.8) | |
| High fall (≥ 1m) | 16 (41.0) | 110 (46.2) | |
| Falling objects | 2 (5.1) | 21 (8.8) | |
| Assault | 0 (0) | 1 (0.4) | |
| ISS | | | 0.180 |
| ≤15 | 7 (17.9) | 34 (14.3) | |
| 16-25 | 13 (33.3) | 52 (21.8) | |
| >25 | 19 (48.7) | 152 (63.9) | |
| Level of the injury | | | 0.382 |
| Cervical | 35 (89.7) | 193 (81.1) | |
| Thoracic | 2 (5.1) | 29 (12.2) | |
| Lumbar | 2 (5.1) | 16 (6.7) | |
| Sacral | 0 (0) | 0 (0) | |
| The level and severity of the injury | | | 0.177 |
| Complete tetraplegia | 11 (28.2) | 49 (20.6) | |
| Incomplete tetraplegia | 23 (59.0) | 145 (60.9) | |
| Complete paraplegia | 1 (2.6) | 30 (12.6) | |
| Incomplete paraplegia | 4 (10.3) | 14 (5.9) | |
| Steroid treatment | | | 0.852 |
| Yes | 33 (84.6) | 198 (83.2) | |
| No | 6 (15.4) | 40 (16.8) | |
| Mechanical ventilation | | | 0.326 |
| Yes | 18 (46.2) | 130 (54.6) | |
| No | 21 (53.8) | 108 (45.4) | |
| Tracheotomy | | | 0.324 |
| Yes | 10 (25.6) | 80 (33.6) | |

*(Continued)*

**Table 6.** (Continued)

| Characters | Improvement Group N = 39 (14.1%) | Non-Improvement Group N = 238 (85.9%) | P Value |
|---|---|---|---|
| No | 29 (74.4) | 158 (66.4) | |
| VTE | | | 0.820 |
| Yes | 11 (27.2) | 63 (26.5) | |
| No | 28 (71.8) | 175 (73.5) | |
| Pneumonia | | | 0.998 |
| Yes | 20 (51.3) | 122 (51.3) | |
| No | 19 (48.7) | 116 (48.7) | |
| The use of mannitol | | | 0.304 |
| Yes | 34 (87.2) | 191 (80.3) | |
| No | 5 (12.8) | 47 (19.7) | |
| Early targeted blood pressure management | | | 0.041 |
| Yes | 29 (74.4) | 139 (58.4) | |
| No | 10 (25.6) | 99 (41.6) | |
| Early surgery (≤24 h) | | | 0.014 |
| Yes | 8 (20.5) | 19 (8.0) | |
| No | 31 (79.5) | 219 (92.0) | |
| Surgical approach | | | 0.878 |
| Anterior | 18 (46.2) | 109 (45.8) | |
| Posterior | 17 (43.6) | 110 (46.2) | |
| Middle | 4 (10.3) | 19 (8.0) | |

**Table 7.** Multivariate logistic regression analysis of postoperative neurological improvement in TSCI patients (Adjusted for age, gender, ISS, injury level/severity, steroid use, mannitol use, combined injury, and surgical approach).

| Characters | B | SE | Wald | OR | 95%CI | P |
|---|---|---|---|---|---|---|
| Early target blood pressure | 0.831 | 0.406 | 2.049 | 2.296 | (1.036, 5.086) | 0.040 |
| Early surgery | 1.044 | 0.490 | 2.132 | 2.841 | (1.088, 7.419) | 0.033 |
| Age | 0.033 | 0.208 | 0.157 | 1.033 | (0.687, 1.554) | 0.875 |
| Gender | 0.604 | 0.458 | 1.319 | 1.830 | (0.746, 4.492) | 0.187 |
| ISS | −0.321 | 0.234 | −1.369 | 0.725 | (0.459, 1.149) | 0.171 |
| Combined injury | −2.502 | 1.577 | −1.586 | 0.082 | (0.004, 1.802) | 0.113 |
| The level and severity of the injury | −0.138 | 0.281 | −0.490 | 0.871 | (0.501, 1.511) | 0.624 |
| Steroid treatment | −0.272 | 0.535 | −0.507 | 0.762 | (0.267, 2.177) | 0.612 |
| The use of mannitol | 0.440 | 0.561 | 0.785 | 1.553 | (0.517, 4.662) | 0.432 |
| Surgical approach | 0.057 | 0.277 | 0.207 | 1.059 | (0.615, 1.821) | 0.836 |

During the 5-years study period, the average age at injury rose modestly from 53.3 to 56.0 years. The most commonly affected age group was 46−60 years, consistent with findings from other cities in China such as Chongqing [16]. However, epidemiological patterns vary across regions: in northwest China, the most affected group was 21–40 years old, whereas in Wuhan, the mean age of TSCI patients was 39.4 years [20]. These discrepancies may reflect regional differences in socioeconomic development, occupational structure, healthcare access, and the impact of COVID-19 restrictions.

Occupationally, peasants (37.8%) and laborers (35.9%) constituted the majority of TSCI cases, corroborating findings from other domestic studies [16,17]. These populations often engage in high-risk, low-protection labor environments.

Additionally, most patients had a relatively low level of education, which may contribute to their limited access to injury prevention knowledge and safer job opportunities.

Neurological recovery remains the primary determinant of prognosis in TSCI patients. A systematic review of 51 studies found that only 10–15% of patients with complete spinal cord injuries regain partial neurological function and transition to an incomplete state [22]. In our cohort, there was no significant difference in neurological improvement rates between surgical (13.0%) and conservative (9.4%) treatments, suggesting that short-term treatment efficacy remains limited in ICU-admitted patients. While preclinical studies have demonstrated the neuroprotective effects of corticosteroids, clinical results are inconsistent, and high-dose steroid regimens are associated with increased risks of infection, sepsis, and respiratory complications [23–30]. Consequently, high-dose methylprednisolone is no longer recommended in current guidelines [31,32]. In our study, 82.4% of patients received standard-dose steroid therapy, and none received high-dose regimens, aligning with evidence-based recommendations [32]. Hemodynamic optimization—specifically, maintaining a MAP ≥ 85 mmHg during the acute post-injury phase—has been strongly recommended to improve outcomes [33]. Our multivariate analysis supports this approach, showing that early MAP maintenance was significantly associated with better neurological recovery (OR=2.233, 95% CI: 1.025–4.866, P = 0.043). It should be noted, however, that this study did not include follow-up analysis of the long-term prognosis of patients, which may have led to an underestimation of the actual treatment effects. Patients received different steroid dosages based on clinical judgment, but no patient received high-dose steroid pulse therapy, and the specific impact of these dosing differences on neurological outcomes requires further investigation.

Timely surgical decompression is another well-established strategy for improving neurological outcomes. Evidence suggests that early surgery (≤ 24h) reduces secondary injury and enhances recovery [34,35]. Based on these findings, related guidelines recommend early surgery as an effective therapeutic approach [33,36]. The guidelines advise performing decompression and spinal stabilization surgery within 24 hours for patients with severe neurological deficits, whether the injuries are complete or incomplete, provided life is not endangered [19,33]. However, in this study, only 8.6% of traumatic SCI patients underwent surgery within 24 hours. This low percentage may be attributed to the severe and complex conditions of traumatic SCI patients admitted to the ICU, leading to surgical delays and potentially affecting surgical outcomes.

To further explore the key factors affecting neurological recovery in ICU patients, particularly the impact of early (≤ 24h) versus non-early (> 24h) surgeries, logistic regression analysis was performed on postoperative neurological improvement. The analysis revealed that surgical timing significantly influences neurological recovery in TSCI patients. Patients undergoing early (≤ 24h) surgery had 2.841 times higher odds of neurological improvement compared to those receiving non-early surgery. These findings emphasize the importance of guideline-recommended early decompression and spinal stabilization surgeries [19,36,37]. Early surgical intervention may mitigate secondary spinal cord injury caused by sustained compression, thereby enhancing the likelihood of neurological recovery [19]. Additionally, early surgery may reduce neuroinflammatory responses and cell death, creating a more favorable biological environment for neural regeneration and functional recovery [36,37]. According to the latest 2024 guidelines from the World Society of Emergency Surgery and the European Neurosurgical Society, urgent interventions such as decompression and spinal stabilization surgery should be prioritized for all salvageable traumatic SCI patients with multiple injuries after life-threatening conditions are controlled. If feasible, these surgeries are strongly recommended within 24 hours after trauma (agreement: 92%, strong recommendation) [32]. Although this study supports the importance of early surgery, the proportion of patients undergoing surgery within 24 hours remains low in clinical practice. This likely reflects practical challenges, such as surgical resource availability, delays in transferring patients to appropriate facilities, and the need to stabilize ICU patients before surgery. For patients unable to undergo surgery within 24 hours due to factors such as transfer delays or preoperative evaluation, surgery within 3 days is advised whenever feasible [38,39]. Medical professionals should continue to explore ways to perform surgery earlier while ensuring patient safety and maintaining MAP ≥ 85 mmHg to promote better neurological recovery.

LOS is an important indicator for assessing patient costs, particularly for TSCI patients [40]. In our study, LOS in both ICU and hospital settings was significantly reduced in the post-pandemic control group. Additionally, previous studies have shown that patients with complications following TSCI tend to have longer LOS, especially those with pneumonia [41,42]. In this study, the incidence of pneumonia and the need for mechanical ventilation were significantly reduced in the post-pandemic control group compared to the pre-pandemic group. Furthermore, the time to initiate surgery after injury was shorter in the post-pandemic period. Early surgical treatment not only facilitates overall recovery but also helps reduce the length of hospital stay [19]. The implementation of COVID-19 measure policies, including restricted family visits, may have encouraged earlier discharge or transfers to local facilities. Furthermore, the launch of Diagnosis-Related Group (DRG)-based reimbursement policies in Zhejiang Province in 2020 likely incentivized shorter LOS [43].

This study has several limitations: First, the ASIA Impairment Scale, while widely used, lacks anatomical specificity and does not capture non-neurological outcomes such as pain, spasticity, or autonomic dysfunction [44–46]. Standardized assessments such as the Spinal Cord Independence Measure (SCIM) or the International Standards to document remaining Autonomic Function after Spinal Cord Injury (ISAFSCI) were not applied, limiting functional outcome evaluation [47–49]. Second, the small number of patients in the improvement group (n = 39) may reduce statistical power. Third, while patients with craniocerebral injuries were included to reflect real-world ICU populations, this could confound neurological outcome assessments. Fourth, the exclusion of ASIA grade E patients may underestimate rare neurological deterioration events. Future studies should address these gaps and include long-term follow-up data to assess functional recovery more accurately. Fifth, our study did not include a detailed spinal fracture morphology radiographic classification. This limits our ability to evaluate the relationship between specific fracture types and neurological outcomes. Given that fracture patterns may influence the mechanism and severity of spinal cord injury, future studies should incorporate standardized imaging assessments to explore how different morphologies correlate with recovery trajectories.

## Conclusion

Patients with TSCI admitted to the ICU are predominantly middle-aged men, with high falls and traffic accidents being the primary etiologies. Incomplete quadriplegia due to cervical spinal cord injury is the most common injury pattern, typically accompanied by a high prevalence of associated injuries and complications. Despite the severity of these cases, in-hospital mortality remained low. Early targeted blood pressure management (MAP ≥ 85 mmHg) and prompt surgical intervention (≤ 24h) were both independently associated with improved neurological recovery, underscoring their importance in the acute management of TSCI. However, the short-term neurological improvement rate remains unsatisfactory, highlighting the need for enhanced treatment protocols and long-term rehabilitation strategies.

## Author contributions

**Conceptualization:** Danqin Yuan.

**Data curation:** Danqin Yuan, Min Tang.

**Formal analysis:** Danqin Yuan, Yaling Jin, Jiuzhou Lin.

**Investigation:** Yaling Jin.

**Methodology:** Yaling Jin, Lihui Chen.

**Project administration:** Weiting Chen.

**Resources:** Lihui Chen.

**Software:** Lihui Chen, Min Tang, Jiuzhou Lin.

**Validation:** Jiuzhou Lin.

**Writing – original draft:** Danqin Yuan.

**Writing – review & editing:** Weiting Chen.

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
