## [Decision Letter · Decision Letter 0]

18 Mar 2025

PONE-D-25-08324Neurological Outcomes and Predictive Factors in Traumatic Spinal Cord Injury Patients in the Intensive Care UnitPLOS ONE

Dear Dr. Chen,

Thank you for submitting your manuscript to PLOS ONE. After careful consideration, we feel that it has merit but does not fully meet PLOS ONE’s publication criteria as it currently stands. Therefore, we invite you to submit a revised version of the manuscript that addresses the points raised during the review process.

We look forward to receiving your revised manuscript.

Kind regards,

Osama Farouk

Academic Editor

PLOS ONE

Reviewers' comments:

Reviewer's Responses to Questions

**Comments to the Author**

1. Is the manuscript technically sound, and do the data support the conclusions?

Reviewer #1: Partly

Reviewer #2: Yes

2. Has the statistical analysis been performed appropriately and rigorously? 

Reviewer #1: No

Reviewer #2: Yes

3. Have the authors made all data underlying the findings in their manuscript fully available?

Reviewer #1: Yes

Reviewer #2: Yes

4. Is the manuscript presented in an intelligible fashion and written in standard English?

Reviewer #1: Yes

Reviewer #2: Yes

5. Review Comments to the Author

Reviewer #1: Thanks for the opportunity to review this manuscript.

major comments:

1. Depending on the ASIA scale to determine neurological recovery is depatable, so the authors should mention in the abstract that they used this scale as a method of measurement of neurological improvement.

2. A limitation section should be written to address the following issues: 1: using ASIA scale as a measure of functional recovery. 2: the limited number of patients in the improvement group (39) compared to no improvement (236). 3: including patients with craniocerebral injury is a significant bias as the authors can include patients with TBI that should have impacts on functional recovery as well.

3. The authors mentioned using multivariate analysis, while the presented tables included only univariate and logistic regression analysis. I cannot trust the results based on univariate analysis in a heterogenous problem like SCI. The authors should perform multivariate analysis to adjust for age, gender, ASIA scale, and combined injuries to see the exact impact of blood pressure management and early surgery on neurological recovery.

minor comments:

1. In Table 6, please revise the number of patients without improvement throughout all factors. The authors said in the first row that the number is 236 (non-improvement group), while for example, at early surgery and after adding patients with and without early surgery, the total is 238.

Reviewer #2: I have reviewed this interesting manuscript and have the following questions and comments:

The authors stated that “Data was accessed for research purposes between 12/4/2023 and 12/4/2025” putting in mind that we are still in march 2025!!??

Why were patient with grade E ASIA scale excluded from the study putting in mind the possibility of deterioration along the course of treatment?

Why were some patient given high dose steroids and others given regular dose and how did this impact the results?

6. PLOS authors have the option to publish the peer review history of their article (what does this mean? ). If published, this will include your full peer review and any attached files.

**Do you want your identity to be public for this peer review?** For information about this choice, including consent withdrawal, please see our Privacy Policy .

Reviewer #1: No

Reviewer #2: No

---

## [Author Response · Author response to Decision Letter 1]

20 Mar 2025

Response to Reviewers

Manuscript ID: PONE-D-25-08324

Title: Neurological Outcomes and Predictive Factors in Traumatic Spinal Cord Injury Patients in the Intensive Care Unit

Dear Academic Editor and Reviewers,

Thank you very much for your valuable comments and insightful suggestions on our manuscript. We have carefully revised our manuscript according to your feedback. Below, we provide point-by-point responses to each comment:

Reviewer #1:

Major comments:

1. Depending on the ASIA scale to determine neurological recovery is debatable. Authors should mention clearly in the abstract that they used this scale as a method of measurement of neurological improvement.

Response: We appreciate your point regarding the debate surrounding the ASIA scale as a measure of neurological recovery. To clarify our methodology, we have revised the abstract to explicitly state that the ASIA scale was used to assess neurological improvement (see revised Abstract). Additionally, we have added a discussion of its strengths and limitations in the newly included Limitations section (see Discussion).

2. A limitation section should address these issues clearly:

a. Using ASIA scale as a measure of functional recovery.

b. The limited number of patients in the improvement group compared to the non-improvement group.

c. Including patients with craniocerebral injury.

Response: A clear limitation section has been added to address these points explicitly:

"This study has several limitations: First, the ASIA scale, though commonly used, may not comprehensively reflect functional recovery. Second, the relatively small number of patients in the improvement group (n=39) compared to the non-improvement group (n=238) may affect statistical robustness. Third, The inclusion of patients with craniocerebral injuries as a potential source of bias, with an explanation that these patients were included to reflect real-world ICU populations, though we now discuss how traumatic brain injury (TBI) could confound functional outcomes." (Discussion, Limitations paragraph)

3. Authors mentioned using multivariate analysis, but the presented tables included only univariate and logistic regression analysis. Perform multivariate analysis adjusting for age, gender, ASIA scale, and combined injuries.

Response: We have clarified our statistical methods and performed multivariate logistic regression analyses, clearly adjusting for age, gender, ISS, The level and severity of the injury, Steroid treatment, The use of mannitol, combined injuries and Surgical approach. The revised methods section states:

"Univariate and multivariate logistic regression analyses were conducted to identify significant factors. Multivariate analyses adjusted for age, gender, ISS, The level and severity of the injury, Steroid treatment, The use of mannitol, combined injuries and Surgical approach were conducted to evaluate the impact on neurological recovery." (Methods section)

Results from adjusted analyses have been updated in Table 7.

Minor comments:

1. In Table 6, revise the number of patients without improvement (from 238 to 236).

Response: The number of patients has been corrected from 236 to 238 in Table 6. This was a typographical error, and we appreciate your attention to detail.

Reviewer #2:

1. Clarify the correct period for data access.

Response: We have corrected the data access date to read: "Data was accessed for research purposes between December 4, 2023, and December 4, 2024." (Methods section)

2. Explain why ASIA Grade E patients were excluded.

Response: We have clarified the exclusion reason: "Patients classified as ASIA grade E were excluded due to the lack of neurological deficits, with minimal risk of functional deterioration significant enough for inclusion in this study." However, we acknowledge your concern about potential deterioration. We have clarified this rationale in the Methods section and added a note in the Limitations section discussing the possibility of deterioration and its exclusion from our primary outcome.

3. Clarify why patients were given different steroid dosages and how this might impact results.

Response: We clarified this issue in the discussion: "Patients received different steroid dosages based on clinical judgment, but no patient received high-dose steroid pulse therapy, and the specific impact of these dosing differences on neurological outcomes requires further investigation." (Discussion section)

Additional Revisions per Journal Requirements:

1.PLOS ONE style requirements:

We have reviewed and adjusted the manuscript to comply with PLOS ONE’s formatting guidelines, including file naming conventions.

2.Data Availability Statement:

We have updated the submission form with the statement: “All data are in the manuscript and/or supporting information files,” as all relevant data are included in the submission.

3.Ethics statement placement:

The ethics statement has been moved to the Methods section and removed from other sections as requested.

Thank you again for your valuable feedback. We hope our revisions adequately address your concerns and look forward to your further evaluation.

Sincerely,

Weiting Chen, M.D.

Corresponding author

---

## [Decision Letter · Decision Letter 1]

15 Apr 2025

PONE-D-25-08324R1Neurological Outcomes and Predictive Factors in Traumatic Spinal Cord Injury Patients in the Intensive Care UnitPLOS ONE

Dear Dr. Chen,

Thank you for submitting your manuscript to PLOS ONE. After careful consideration, we feel that it has merit but does not fully meet PLOS ONE’s publication criteria as it currently stands. Therefore, we invite you to submit a revised version of the manuscript that addresses the points raised during the review process.

We look forward to receiving your revised manuscript.

Kind regards,

Osama Farouk

Academic Editor

PLOS ONE

Journal Requirements:

Additional Editor Comments (if provided):

Reviewers' comments:

Reviewer's Responses to Questions

**Comments to the Author**

1. If the authors have adequately addressed your comments raised in a previous round of review and you feel that this manuscript is now acceptable for publication, you may indicate that here to bypass the “Comments to the Author” section, enter your conflict of interest statement in the “Confidential to Editor” section, and submit your "Accept" recommendation.

Reviewer #1: All comments have been addressed

Reviewer #2: All comments have been addressed

Reviewer #3: (No Response)

2. Is the manuscript technically sound, and do the data support the conclusions?

Reviewer #1: Yes

Reviewer #2: Yes

Reviewer #3: (No Response)

3. Has the statistical analysis been performed appropriately and rigorously? 

Reviewer #1: Yes

Reviewer #2: Yes

Reviewer #3: (No Response)

4. Have the authors made all data underlying the findings in their manuscript fully available?

Reviewer #1: Yes

Reviewer #2: Yes

Reviewer #3: (No Response)

5. Is the manuscript presented in an intelligible fashion and written in standard English?

Reviewer #1: Yes

Reviewer #2: Yes

Reviewer #3: (No Response)

6. Review Comments to the Author

Reviewer #1: - Thanks for addressing the previous comments. Please consider also the following suggestions:

1. Add the effect size for your multivariate analysis so we can determine how much stronger that early surgery and blood pressure management can be to determine neurological recovery.

2. Can you adjust the title of Table 7 to clearly identify your adjustment?

3. For your limitation section, can you add references and elaborate more on the limitations of the ASIA scale?

4. Did you consider any questionnaire for autonomic recovery as well? It will be important to elaborate on this.

5. From your record, what is the shape of spinal fracture that was associated with the best neurological recovery, and what was the type of spinal fracture that was associated with the worst recovery?

6. Please clearly identify in the abstract that you used the ASIA scale as a measure of neurological recovery.

Reviewer #2: (No Response)

Reviewer #3: All looks well statistically. The revisions have been addressed and additions to the paper included, especially with the multivariate logistic addition.

Some typos noticed and important: Authors abstract:

Result section: 'Multivariate logistic regression analysis demonstrated that early targeted blood pressure management (MAP≥ 85mmHg) (OR=2.296, 95% CI: 1.036–5.086, P=0.040) and early surgery (≥24h) (OR=2.841, 95% CI: 1.088–7.419, P=0.033) were protective factors for neurological improvement.'

Correction,

‘and early surgery (≥24h)’ should be,’ and early surgery (<= 24h).’

Also 'Treatment of TSCI and Status at Discharge' section ,

Correction,

At very end two lines before the ‘Discussion’ section

‘and early surgery (≥24h)’ should be ‘and early surgery (<=24h’)'.

Please check and edit any other typos in the paper.

7. PLOS authors have the option to publish the peer review history of their article (what does this mean? ). If published, this will include your full peer review and any attached files.

**Do you want your identity to be public for this peer review?** For information about this choice, including consent withdrawal, please see our Privacy Policy .

Reviewer #1: No

Reviewer #2: No

Reviewer #3: No

---

## [Author Response · Author response to Decision Letter 2]

17 Apr 2025

Response to Reviewers

Manuscript ID: PONE-D-25-08324R1

Title: Neurological Outcomes and Predictive Factors in Traumatic Spinal Cord Injury Patients in the Intensive Care Unit

Dear Academic Editor and Reviewers,

We sincerely appreciate the reviewers' positive feedback and the opportunity to further improve our manuscript. We have carefully addressed all the comments and suggestions as outlined below:

Reviewer #1:

1.Add the effect size for your multivariate analysis so we can determine how much stronger that early surgery and blood pressure management can be to determine neurological recovery.

Response: We have added effect size data in the Results section and in Table 7, including standard errors (SE) for each covariate. Specifically, we report the odds ratios (OR), 95% confidence intervals (CI), and SEs from the multivariate logistic regression model, which demonstrate the strength of association between early surgical intervention and targeted blood pressure management. These findings suggest that both early hemodynamic optimization and timely surgical intervention are independently associated with improved neurological recovery. For instance, early surgery (OR = 2.841, 95% CI: 1.088–7.419, SE = 0.490) and early blood pressure management (OR = 2.296, 95% CI: 1.036–5.086, SE = 0.406) were found to be significant predictors of favorable neurological outcomes.

2.Can you adjust the title of Table 7 to clearly identify your adjustment?

Response: We have revised the title of Table 7 to: "Multivariate Logistic Regression Analysis of Postoperative Neurological Improvement in Traumatic SCI Patients (Adjusted for age, gender, ISS, injury level/severity, steroid use, mannitol use, combined injury, and surgical approach)."

3.For your limitation section, can you add references and elaborate more on the limitations of the ASIA scale?

Response: We have expanded the Limitations section to include a more detailed discussion of the limitations of the ASIA scale and added relevant references to support these points. We also clarified in the Methods section that patients classified as ASIA grade E were excluded due to the lack of neurological deficits, with minimal risk of functional deterioration significant enough for inclusion. However, we acknowledge the concern that some of these patients could deteriorate, and this limitation has now been explicitly noted in the revised Limitations section.

4.Did you consider any questionnaire for autonomic recovery as well? It will be important to elaborate on this.

Response: We acknowledge this important aspect. We have now added a sentence in the Discussion noting that autonomic recovery was not assessed using standardized questionnaires such as the Spinal Cord Independence Measure (SCIM)  or the International Standards to document remaining Autonomic Function after Spinal Cord Injury (ISAFSCI) , which is a limitation of our study and should be addressed in future research.

5.From your record, what is the shape of spinal fracture that was associated with the best neurological recovery, and what was the type of spinal fracture that was associated with the worst recovery?

Response: We appreciate the reviewer’s interest in the association between spinal fracture morphology and neurological recovery. However, our current dataset did not include sufficient detail on fracture shape classification to allow for a reliable analysis of fracture patterns and their correlation with outcomes. Therefore, we are unable to provide conclusive observations regarding which fracture types were associated with the best or worst neurological recovery. We have clarified this in the Discussion section and suggested it as an important area for future research, which we plan to investigate in subsequent studies with dedicated radiographic data collection and classification. We are particularly interested in exploring how specific fracture morphologies may influence spinal cord damage and recovery trajectories in different clinical scenarios.

6.Please clearly identify in the abstract that you used the ASIA scale as a measure of neurological recovery.

Response: This has been clearly stated in the Methods portion of the Abstract: “Neurological recovery was assessed using the American Spinal Injury Association (ASIA) impairment scale.”

Reviewer #2: (No additional comments provided.)

Reviewer #3:

1.Abstract Result section: 'early surgery (≥24h)' should be corrected to 'early surgery (≤24h).'

Response: Thank you for identifying this. We have corrected this typo in both the Abstract and the main Results section.

2.Also in the 'Treatment of TSCI and Status at Discharge' section, ‘early surgery (≥24h)’ should be ‘early surgery (≤24h).’

Response: This error has been corrected.

3.Please check and edit any other typos in the paper.

Response: We have conducted a thorough proofreading of the entire manuscript and corrected typographical and grammatical errors to improve readability and precision.

We appreciate the thoughtful and constructive feedback from the reviewers and editor.

We believe that the revised manuscript has been significantly improved and hope it now meets the criteria for publication in PLOS ONE.

Sincerely,

Weiting Chen, M.D. Corresponding Author

---

## [Decision Letter · Decision Letter 2]

8 May 2025

Neurological Outcomes and Predictive Factors in Traumatic Spinal Cord Injury Patients in the Intensive Care Unit

PONE-D-25-08324R2

Dear Dr. Chen,

We’re pleased to inform you that your manuscript has been judged scientifically suitable for publication and will be formally accepted for publication once it meets all outstanding technical requirements.

Kind regards,

Osama Farouk

Academic Editor

PLOS ONE

Additional Editor Comments (optional):

Reviewers' comments:

Reviewer's Responses to Questions

**Comments to the Author**

1. If the authors have adequately addressed your comments raised in a previous round of review and you feel that this manuscript is now acceptable for publication, you may indicate that here to bypass the “Comments to the Author” section, enter your conflict of interest statement in the “Confidential to Editor” section, and submit your "Accept" recommendation.

Reviewer #1: All comments have been addressed

Reviewer #3: All comments have been addressed

2. Is the manuscript technically sound, and do the data support the conclusions?

Reviewer #1: Yes

Reviewer #3: (No Response)

3. Has the statistical analysis been performed appropriately and rigorously? 

Reviewer #1: Yes

Reviewer #3: (No Response)

4. Have the authors made all data underlying the findings in their manuscript fully available?

Reviewer #1: Yes

Reviewer #3: (No Response)

5. Is the manuscript presented in an intelligible fashion and written in standard English?

Reviewer #1: Yes

Reviewer #3: (No Response)

6. Review Comments to the Author

Reviewer #1: Thanks a lot for the opportunity to review this paper.

The authors addressed my comments appropriately.

Reviewer #3: (No Response)

7. PLOS authors have the option to publish the peer review history of their article (what does this mean? ). If published, this will include your full peer review and any attached files.

**Do you want your identity to be public for this peer review?** For information about this choice, including consent withdrawal, please see our Privacy Policy .

Reviewer #1: No

Reviewer #3: No

---

## [Editor Report · Acceptance letter]

PONE-D-25-08324R2

PLOS ONE

Dear Dr. Chen,

I'm pleased to inform you that your manuscript has been deemed suitable for publication in PLOS ONE. Congratulations! Your manuscript is now being handed over to our production team.

Kind regards,

on behalf of

Dr. Osama Farouk

Academic Editor

PLOS ONE